# New, Biocompatible, Chitosan-Gelled Microemulsions Based on Essential Oils and Sucrose Esters as Nanocarriers for Topical Delivery of Fluconazole

**DOI:** 10.3390/pharmaceutics14010075

**Published:** 2021-12-29

**Authors:** Lavinia Vlaia, Ioana Olariu, Ana Maria Muţ, Georgeta Coneac, Vicenţiu Vlaia, Dan Florin Anghel, Monica Elisabeta Maxim, Gabriela Stângă, Amadeus Dobrescu, Maria Suciu, Zoltan Szabadai, Dumitru Lupuleasa

**Affiliations:** 1Department II—Pharmaceutical Technology, Formulation and Technology of Drugs Research Center, “Victor Babeș” University of Medicine and Pharmacy, 300041 Timișoara, Romania; vlaia.lavinia@umft.ro (L.V.); olariu.ioana@umft.ro (I.O.); mut.anamaria@umft.ro (A.M.M.); coneac.georgeta@umft.ro (G.C.); 2Department I—Organic Chemistry, Formulation and Technology of Drugs Research Center, “Victor Babeș” University of Medicine and Pharmacy, 300041 Timișoara, Romania; 3“Ilie Murgulescu” Institute of Physical Chemistry of the Romanian Academy, Laboratory of Colloid Chemistry, 060021 Bucharest, Romania; danflorin.anghel@gmail.com (D.F.A.); monimaxim@gmail.com (M.E.M.); gstinga@gmail.com (G.S.); 4Department X Surgery 2–Surgery 2, “Victor Babeș” University of Medicine and Pharmacy, 300041 Timișoara, Romania; dobrescu.amadeus@umft.ro; 5Department II—Pharmacology and Pharmacotherapy, “Victor Babeș” University of Medicine and Pharmacy, 300041 Timișoara, Romania; suciu.maria@umft.ro; 6National Institute for Research and Development in Electrochemistry and Condensed Matter, 300569 Timişoara, Romania; szabadai@umft.ro; 7Department of Pharmaceutical Technology and Biopharmaceutics, Faculty of Pharmacy, “Carol Davila” University of Medicine and Pharmacy, 020956 Bucharest, Romania; dumitru.lupuliasa@umfcd.ro

**Keywords:** biocompatible sucrose ester, essential oil, chitosan, microemulsion, antifungal, fluconazole

## Abstract

Biocompatible gel microemulsions containing natural origin excipients are promising nanocarrier systems for the safe and effective topical application of hydrophobic drugs, including antifungals. Recently, to improve fluconazole skin permeation, tolerability and therapeutic efficacy, we developed topical biocompatible microemulsions based on cinnamon, oregano or clove essential oil (CIN, ORG or CLV) as the oil phase and sucrose laurate (D1216) or sucrose palmitate (D1616) as surfactants, excipients also possessing intrinsic antifungal activity. To follow up this research, this study aimed to improve the adhesiveness of respective fluconazole microemulsions using chitosan (a biopolymer with intrinsic antifungal activity) as gellator and to evaluate the formulation variables’ effect (composition and concentration of essential oil, sucrose ester structure) on the gel microemulsions’ (MEGELs) properties. All MEGELs were evaluated for drug content, pH, rheological behavior, viscosity, spreadability, in vitro drug release and skin permeation and antifungal activity. The results showed that formulation variables determined distinctive changes in the MEGELs’ properties, which were nevertheless in accordance with official requirements for semisolid preparations. The highest flux and release rate values and large diameters of the fungal growth inhibition zone were produced by formulations MEGEL-FZ-D1616-CIN 10%, MEGEL-FZ-D1216-CIN 10% and MEGEL-FZ-D1616-ORG 10%. In conclusion, these MEGELs were demonstrated to be effective platforms for fluconazole topical delivery.

## 1. Introduction

Nowadays, within the group of the most commonly encountered forms of infection, mycoses are the fourth ranked and, among them, superficial and skin fungal infections are the most frequent and widespread forms [1,2]. In the case of superficial and cutaneous fungal infections, represented mainly by candidiasis and dermatophytosis, topical treatment is the first therapy option for mild or localized forms or those spread on limited areas [3,4,5]. An ideal compound for topical treatment of skin fungal infections should have the following characteristics: a broad spectrum of action, high efficacy (mycological cure), a convenient dosing regimen (once- or twice-daily regimen), low incidence of side effects and be relatively inexpensive [6].

Fluconazole (FZ), a first-generation triazole fluorinated derivative, is one of the commonly used synthetic azole compounds and could be an appropriate drug of choice for topical therapy of skin mycoses as it possesses several of these required features: a broad spectrum of action which includes numerous species of *Candida* (i.e., *C. albicans* and *C. neoformans*) and dermatophytes; low toxicity due to its higher affinity for fungal cytocrome P450 enzymes inhibiting ergosterol biosynthesis, in comparison with imidazoles (ketoconazole and miconazole); and high efficacy achieved by fungistatic effect due to ergosterol depletion on the fungal cell membrane [3,7]. On the other hand, in oral therapy of skin mycoses, it was demonstrated that, compared with other azoles (itraconazole and ketoconazole) used in oral therapy of skin mycoses, FZ shows much higher affinity to the skin keratin and, consequently, it accumulates rapidly and greatly in the stratum corneum as an active, nonbinding form after systemic administration. This pharmacokinetic property of FZ largely accounts for its in vivo efficacy [8]. However, after topical administration, the accumulation of FZ or other azoles in the skin in effective concentration levels is limited by the outermost skin layer, the stratum corneum. Hence, to overcome the formidable skin barrier represented by the stratum corneum and to reach subjacent skin layers (especially the viable epidermis), an antimycotic agent must have several specific physical properties, including low molecular weight (<500 Da), electrical neutrality and solubility in both the aqueous and lipid phases [9]. Due to the two triazole rings in its structure, FZ has a moderate lipophilicity indicated by the log *p* of 0.5, which is quite low compared to that of all other azole antifungal agents (i.e., 3.5 for ketoconazole, 5.7 for itraconazole, 1.72 for voriconazole, 5.4 for posaconazole) [10]. Consequently, among the antifungal azoles, FZ has the highest water solubility (1 mg/mL), but, even so, it is considered a slightly water-soluble drug which could be formulated as topical dosage forms containing various solubilizing systems. Nevertheless, because FZ is moderately lipophilic in nature, the main disadvantage of its cutaneous administration is its poor penetration through the stratum corneum to reach the deeper skin layers, particularly the viable epidermis, in order to achieve high efficacy. Therefore, in the last two decades, the extensive investigations to improve FZ skin permeation were focused mainly on two directions: (1) the use of various chemical penetration enhancers incorporated in conventional topical dosage forms (emulsions, hydrogels, oleogels, emulgels and creams) [11,12,13,14,15,16,17]; and (2) the formulation of different nanocarrier systems such as vesicular nanocarriers (liposomes, ethosomes, oleic acid vesicles, nanostructure lipid carriers) and microemulsions, usually loaded in gel vehicles [18,19,20,21,22,23,24,25].

As a part of both above-mentioned strategies for enhancing FZ skin penetration, the use of pharmaceutical excipients of natural origin which are biocompatible, biodegradable and toxicologically harmless can be helpful to achieve important goals for the formulation of topical products: by improving the cutaneous efficacy, safety and tolerability profile of the final product [26,27].

Essential oils (EOs) were included into the new class of natural skin penetration enhancers because they are not only highly potent (their capability to increase the skin penetration of drugs from topical products is based on consistent scientific evidence) [28,29,30], but also safe, falling into the list of generally recognized as safe (GRAS) agents published by the U.S. Food and Drug Administration [31]. Moreover, many published works have demonstrated that essential oils, as complex mixtures of numerous volatile and non-volatile compounds, possess a broad antimicrobial activity against various microorganisms, including fungi, bacteria, protozoaires and viruses [32,33,34]. In addition, some recent studies evaluated the potential synergistic effects of essential oils with synthetic antifungal drugs, such as FZ, in order to exploit the possible synergy in combination therapy which could both counteract antifungal drug resistance and increase treatment efficacy with lower doses of antifungal drug and, consequently, reduce costs and adverse side effects [35,36,37,38].

Alongside essential oils, sucrose fatty acid esters (SEs) are another group of natural-origin, biodegradable, biocompatible and skin-compatible penetration enhancers which have been extensively studied in the last decade for their drug skin permeation enhancement ability [39,40,41,42,43]. Additionally, sucrose esters are non-ionic, non-toxic and non-polluting surfactants commonly used in dermal pharmaceutical formulations to stabilize the emulsion systems and to solubilize the poorly hydrosoluble drugs, acting as emulsifying and solubilizing agents, respectively. These important functions contribute significantly to the enhancement of their effect as skin penetration promoters [44,45]. Moreover, it has been found that sucrose esters exhibit antifungal and antimicrobial activities [46,47], which could potentiate through synergism the effect of antifungal agents in case of their association in dermatological preparations.

Among the non-vesicular nanocarriers, microemulsions (MEs) are mostly proposed for promoting the percutaneous drug permeation. Microemulsions are homogenous, transparent, thermodynamically stable and isotropic colloidal dispersions of two immiscible liquids (oil and water) stabilized by an interfacial monomolecular film of a surfactant or mixed surfactants, usually in combination with a cosurfactant, with mean droplet diameter in the range of 10–200 nm [48]. Numerous recent studies demonstrated that these second-generation colloidal carriers are attractive vehicles for efficient topical drug delivery, exhibiting most of the desired attributes (spontaneous formation and, consequently, preparation with low mechanical energy, long shelf life, transparency related to aesthetic properties, enhanced solubilization capability for both lipophilic and hydrophilic compounds, improved bioavailability and skin penetration enhancement) [49]. Further, several investigations revealed that MEs can provide prolonged drug release, hinder the skin irritation effect of the drug and protect the solubilized (encapsulated) drug from degradation [50,51,52]. However, the required high amounts of surfactant and cosurfactant to obtain MEs could cause cutaneous intolerance, manifested through irritation and sensibilization. However, dermal irritation could be minimalized by careful selection of ME constituents which, in suitable concentrations, have GRAS status (generally regarded as safe) and are clinically accepted for topical formulations [53]. As a group of natural, non-ionic surfactants, sugar fatty acid esters have been frequently investigated in the last decade for their potential to form biocompatible MEs [54,55] due to their low skin irritancy and good environmental compatibility.

In this regard, we have recently published a preliminary study on some microemulsion systems based on sucrose esters surfactants and essential oils. The results of this study demonstrated that the surfactants sucrose laurate and sucrose palmitate, in the presence of isopropyl alcohol as cosurfactant, are able to stabilize L/H microemulsion systems containing a mixture of essential oils (cinnamon, clove or oregano) and isopropyl myristate as the oil phase and water [56]. These systems could be harnessed for developing biocompatible, effective vehicles for topical administration of antifungals. Thus, fluconazole-loaded hydrophilic microemulsions were successfully prepared and physicochemically characterized [56].

Moreover, another category of ME constituents useful for averting their inconvenient fluidity for topical application are the gelling agents. Therefore, when formulating biocompatible, topical MEs, biopolymers are of first choice as gelling agents to prolong the residence time of the formulation on the skin surface.

Chitosan, a natural polysaccharide also known as deacetylchitin or poly-d-glucosamine, is one of the most frequently used biopolymers as a gelling agent in pharmaceutical formulations due to its unique properties, namely biocompatibility, biodegradability, non-toxicity, low allergenicity and mucoadhesivity. Moreover, within the group of natural polysaccharides, chitosan is the only one positively charged, which gives it intrinsic antifungal properties, and recognized as a potential antifungal drug with considerable activity on resistant strains to antimycotic drugs [57,58,59,60,61]. Consequently, the use of chitosan as a gelling agent in the formulation of topical microemulsions containing an antifungal drug can address several issues, including the increase of the ME’s consistency, the enhancement of antifungal drug activity by synergistic effect and the avoidance of drug resistance by its specific mechanism of fungi suppression. However, the inherent mucoadhesive properties of chitosan were not exploited in our work as the investigated formulations were intended for dermal administration.

Therefore, as an extension of our recently published study [56], this research work aimed to exploit the antifungal and gelling potential of biopolymer chitosan in combination with FZ and other components with intrinsic antimycotic effect, namely an essential oil (cinnamon essential oil, oregano essential oil or clove essential oil) and a sucrose ester (laurate or palmitate) for developing biocompatible, hydrophilic gel microemulsions suitable for topical administration and with local antifungal effect improved by a synergistic mechanism solving drug fungal resistance. More precisely, an attempt has been made to increase the viscosity of the previously selected and physicochemically characterized fluconazole-loaded, biocompatible microemulsion formulations using chitosan as a gelling agent. The obtained FZ-loaded gel microemulsions were evaluated for their physicochemical, rheological, in vitro drug release and skin permeation and antifungal properties. The influence of the type and concentration of the essential oil and of the structure of the used sucrose esters on the measured parameters was also investigated.

## 2. Materials and Methods

### 2.1. Materials

Fluconazole (FZ) and chitosan (CTS) with mean molecular weight (300–2000 kDa) (Chitopharm^®^M) were receive as gift samples from Vim Spectrum (Corunca, Romania) and Antibiotice Iaşi (Iaşi, Romania), respectively. Cinnamon essential oil (CIN), clove essential oil (CLV) and oregano essential oil (ORG) were purchased from Elemental (Oradea, Romania). Isopropyl myristate (IPM) was kindly donated by Cognis (Monheim, Germany) and ethyl oleate was purchased from Sigma Aldrich (Taufkirchen, Germany). Natural surfactants, sucrose laurate (D1216) and sucrose palmitate (D1616) were received as free samples from Mitsubishi Chemical Foods Corporation (Tokyo, Japan). Isopropyl alcohol (*IPA*) and ethyl alcohol were provided by Chemical Company (Iaşi, Romania). Acetic acid and sodium chloride were purchased from Chimopar (Bucureşti, Romania). Copper (II) chloride (CuCl_2_·H_2_O) was supplied by Sigma Aldrich (Taufkirchen, Germany). Acetonitrile was purchased from Acros Organics (Geel, Belgium). Distilled water was used to prepare the microemulsions, and bidistilled water was used to prepare the 0.9% isotonic sodium chloride solution. Polysulfone hydrophilic synthetic membranes (Tuffryn HT membranes, 0.45 μm and 25 mm) were supplied by Pall Corporation (Port Washington, NY, USA), and the porcine skin was excised from pork ears purchased from a local slaughterhouse. Except for the porcine skin, all the materials used in the study were of pharmaceutical or analytical purity and were used as received.

### 2.2. Methods

#### 2.2.1. Preparation of Fluconazole-Loaded Microemulsions and Fluconazole-Loaded Gel Microemulsions

Ten fluconazole-loaded, biocompatible microemulsion systems, physicochemically characterized in our previous study [56], were formulated and prepared in the gel form using CTS as gelling agent at a concentration of 1.9% (*w/w*). The corresponding drug-unloaded microemulsion formulations were selected from the phase diagrams of five pseudo-ternary systems (CIN + IPM (1:1 mass ratios)/D1616 + isopropyl alcohol (1:1.5 mass ratios)/water; ORG + IPM (1:1 mass ratios)/D1616 + isopropyl alcohol (1:2 mass ratios)/water; CLV + IPM (1:1 mass ratios)/D1616 + isopropyl alcohol (1:2 mass ratios)/water; CIN + IPM (1:1 mass ratios)/D1216 + isopropyl alcohol (1:2 mass ratios)/water; ORG + IPM (1:1 mass ratios)/D1216 + isopropyl alcohol (1:1.5 mass ratios)/water) based on several criteria, as presented in our previous work [56].

The microemulsions were prepared at room temperature, as described, by first mixing the oil phase (essential oil and IPM) with the surfactant–cosurfactant (SE-IPA) solution, then FZ was dissolved into the respective mixture under moderated magnetic stirring (Hei-Tec hotplate magnetic stirrer, Heidolph, Schwabach, Germany). Finally, the obtained FZ solution was titrated with the proper amount of aqueous phase (2% acetic acid solution), also under magnetic stirring [56].

To obtain the gel microemulsions, CTS was added in small portions, under moderate stirring, to the fluconazole-loaded microemulsions, containing as aqueous phase a 2% acetic acid solution. Stirring was continued until complete dissolution of CTS and homogeneous and transparent gelled microemulsions were obtained. Table 1 and Table 2 present the labels and the composition with the proportion of each constituent of the studied fluconazole-loaded gel microemulsions.

All prepared gelled microemulsions were kept in sealed containers at room temperature for 24 h to equilibrate before further characterization.

#### 2.2.2. Characterization of Chitosan-Gelled Microemulsions Loaded with 2% FZ

(a)Determination of drug content and pH

To measure the FZ content of the chitosan-gelled MEs, 0.4 g of the sample was weighed and quantitatively added to a mixture of acetonitrile–aqueous CuCl_2_ solution (9:1) in a 25 mL volumetric flask. It is to be noted that the concentration of the cupric chloride aqueous solution was 0.53 g/mL. After filtration (0.45 μm Millipore filter membrane), the FZ concentration in the resulting solution was analyzed spectrophotometrically (T70+ spectrophotometer, PG Instruments, Wibtoft, UK) in visible domain, at the wavelength of 800 nm (for cinnamon and clove essential oils) and of 762 nm (for ORG). Each experiment was performed in triplicate at ambient temperature.

The pH values of the aqueous dispersions containing 5% (*w/w*) fluconazole-loaded gel microemulsion were measured at 25 °C using a digital pH meter (SensionTM, Hach Company, Loveland, CO, USA). Each experiment was performed in triplicate.

(b) Rheological measurements

The rheological behavior and viscosity of the fluconazole-loaded gel microemulsions were evaluated by steady shear test performed at 23 °C using a stress-controlled rheometer (RheoStress 1, Thermo Haake, Karlsruhe, Germany) equipped with a cone-plate geometry (C60/1ºTi, 60 mm diameter, 1° cone angle and 0.052 mm gap size). The rheograms and viscosity curves (shear stress and respectively apparent viscosity plotted versus shear rate,) were recorded under controlled rate in a continuous up and down rate ramp experiment: the share rate was progressively increased over a range of 0.05–100 1/s during 120 s and then decreased from the maximum to the minimum value of this range in another 120 s. The resulting flow data were fitted to Ostwald de Waele, Herschel–Bulkley and Casson rheological models, according to respective Equations (1)–(3):(1)$$\tau =K\cdot {\dot{\gamma }}^{n},$$
(2)$$\tau =\tau_{0}+K\cdot {\dot{\gamma }}^{n},$$
(3)$$\tau =\sqrt[{n}]{\tau_{0}^{n}+{\left(\eta_{0}\cdot \dot{\gamma }\right)}^{n}},$$
where *τ* is the shear stress (Pa), *τ*_0_ is the yield stress (Pa), *η* is the viscosity (Pa·s), *K* is the consistency index, $\dot{\gamma }$ is the shear rate (1/s) and *n* is a rheological exponent called flow behavior index (dimensionless). For 0 < *n* < 1, the system exhibits a shear thinning behavior; usually, the smaller the value of *n* is, the more the system is shear thinning. The fitting accuracy was evaluated based on the coefficient of determination value (R^2^). The software Haake RheoWin Data Manager v. 4.3 was used for modelling the samples’ flow behavior and processing all rheological results.

Additionally, the spreadability, a rheological feature nearly related to consistency, was determined at 25 °C by parallel-plate method using the Pozo Ojeda-Sune Arbussa extensometer according to the procedure described in the literature [62]. The spreading surfaces reached by the samples were calculated (mm) and plotted versus the applied weight to obtain the extensometric curves.

All rheological tests were carried out in triplicate.

#### 2.2.3. In Vitro Drug Release and Skin Permeation Studies

(a)In vitro drug release studies

The experiments were performed with a six Franz diffusion cells assembly (Microette-Hanson system, model 57-6AS9, Chatsworth, CA, USA), with an effective diffusion area of 1.767 cm^2^ and a volume of receptor compartment of 6.5 mL. Artificial hydrophilic polysulfone membranes (0.45 μm HT Tuffryn membranes, Pall Corporation, Port Washington, NY, USA) were used. To maintain sink conditions throughout the in vitro test, the acceptor compartment of each diffusion cell was filled with physiological saline solution containing 60% (*v/v*) ethanol, freshly prepared, heated and deaerated. The artificial membranes were fixed between the donor and receptor compartments after being soaked in the acceptor medium for 30 min. An approximate 0.300 g sample was weighed in the donor chamber and placed on the artificial membrane, taking care to avoid air bubble incorporation. Throughout the experiments, the system was maintained at 32 ± 2 °C, and the receptor medium was magnetically stirred at 600 rpm to avoid the effects of the diffusion layer.

The experiments were performed for 8 h. Aliquots of 0.5 mL of the receptor fluid were collected automatically at 0.5, 1, 2, 3, 4, 5, 6, 7 and 8 h and replaced with the same volume of fresh receptor medium to maintain sink conditions. A 0.5 mL amount of 0.0514% (*w*/*v*) copper chloride (II) aqueous solution was added to the samples to form the fluconazole-Cu (II) complex.

The quantitative determination of FZ in the samples was performed by UV spectrophotometric method (UV-VIS T70 + spectrophotometer with UVWIN5 software, PG Instruments, England) at a 330 nm wavelength, corresponding to the maximum absorption of the FZ–Cu (II) complex in normal saline solution containing 60% ethanol. The assay was linear in the FZ concentration range of 48–480 µg/mL (*y* = 0.0074*x* − 0.0071, *R^2^* = 0.9997).

(b) In vitro skin permeation studies

Preparation of skin samples. The in vitro permeation studies were performed using as a model membrane the excised skin from the ears of domestic pigs, 4 months old (female or male), obtained from a local slaughterhouse. Immediately after excision, the pig ears were washed with tap water, then the hair was removed from the outer area of the ears, and the skin was excised using a dermatome (Acculan 3 Ti Electrodermatome, Aesculap—a BBraun Company, Hazelwood, MO, USA) at a thickness of about 500 μm. The excised skin samples were used immediately for permeation tests or kept at −18 °C for up to 2 months. Before use, the excised frozen ear skin was allowed to thaw at room temperature. The integrity of the skin was visually examined and only the samples without physical damage were used in the permeation experiments. The thickness of each skin sample was measured with a micrometer, and then 2 cm^2^ squares were cut from the skin samples.

Skin permeation experiments. To examine the in vitro FZ permeation through pig ear skin, vertical-type diffusion cell experiments were carried out, as described above, but with the following modifications: physiological saline solution as receptor medium; and sampling of the aliquots at preset intervals of 1, 2, 3, 4, 5, 6, 7, 8, 9, 10, 11, 12, 13, 14, 15, 16, 17, 18, 20, 22 and 24 h. The FZ quantification as FZ–Cu (II) complex in the samples was performed by the same UV spectrophotometric assay (330 nm wavelength). The assay was linear in the FZ concentration range of 48–480 µg/mL (*y* = 0.0089*x* − 0.0209, *R*^2^ = 0.9977).

The in vitro diffusion experiments for each chitosan-gelled microemulsion loaded with FZ were repeated five times.

(c) Data analysis of in vitro drug release and permeation studies

By plotting the average cumulative amount of FZ that permeated through the membrane (μg/cm^2^) versus time (*t*, h), the specific permeation parameters, namely the steady state flux (*Jss*, μg/cm^2^/h), the permeability coefficient (*K_p_*, cm/h) and the lag time (*t_L_*, h), were calculated.

For comparison of the obtained FZ release profiles, the experimental data were fitted to four mathematical equations describing the well-known kinetic models, as follows:-Zero order model:
*M_t_* = *M*_0_ + *K*_0_*t*,
(4)

where *M_t_* represents the amount of drug dissolved during time *t*, *M*_0_ represents the initial amount of drug in solution (is usually zero) and *K*_0_ represents the zero-order release constant expressed in units of concentration/time.

-First order model:*logC* = *logC*_0_ − *K*_1_*t*/2.303,
(5)

where *C_0_* is the initial concentration of drug, *K*_1_ is the first-order release constant and *t* is the time.
-Higuchi model:
*M = K_H_t*^1/2^,
(6)
where *M* represents the amount of active substance released at time *t* and *K_H_* is the Higuchi release constant.
-Korsmeyer–Peppas model:
*M_t_/M_∞_ = K_P_t^n^*,
(7)
where *M_t_/M_∞_* represents the ratio of the amount of substance released at time *t*, *K_P_* is the Korsmeyer–Peppas release rate constant and *n* is the diffusion coefficient. In this case, the data corresponding to the release of 60% of the quantity of drug substance were analyzed.

For each used model, the quality of fit was assessed by linear regression analysis, calculating the coefficient of determination (R^2^). The kinetic model generating the highest value of the determination coefficient was considered as the most suitable for describing the FZ release from the studied chitosan-gelled microemulsions.

#### 2.2.4. In Vitro Antifungal Activity Assay

The antifungal activity of the studied chitosan-gelled MEs loaded with FZ was evaluated against *Candida albicans ATCC-90028* strains using the agar plate diffusion method with the modification that the gel samples added to wells made in the agar medium in Petri dishes. Sabouraud agar medium containing chloramphenicol and gentamicin was used as culture medium. The colonies were diluted with 0.85% sodium chloride sterile solution, and the concentration of the obtained suspensions was adjusted to approximately 3 × 10^8^ colony-forming units/mL using a turbidimeter (DEN-1 McFarland Densitometer, Biosan, Riga, Latvia). The plates with Sabouraud agar medium were inoculated with the *C. albicans* suspension, then allowed for 5 min to adsorb the inoculum. In the next step, a well (one for the control gel and the other for the test gel) was made in the middle of each half of the plate by cutting under sterile conditions an 8 mm diameter portion of medium, then each well was filled with the sample. The plates were then incubated at 37 ± 0.1 °C for 48 h. After the incubation period, the inhibition zones diameters (mm) were measured and the mean values of three measurements (±SD) were used to express the antifungal activity of the chitosan-gelled MEs.

#### 2.2.5. Statistical Analysis of Data

The experimental results were presented as the means ± standard deviation (SD). Whenever applicable, their regression and/or statistical analysis were performed using Microsoft Excel software package (2016, Microsoft Corporation, Redmond, WA, USA). In case of statistical analysis, the level of significance was considered as *p* < 0.05.

## 3. Results

### 3.1. Physicochemical Characterization of Chitosan-Gelled Microemulsions Based on Essential Oil and Sucrose Ester

All experimental gel microemulsions were macroscopically homogeneous, translucent, transparent and yellowish with an odor specific to the volatile oils and acetic acid, meeting the official requirements for visual and olfactory properties. Both essential oils and CTS (the gelling agent) contributed to the yellow coloration of the systems.

#### 3.1.1. Determination of Drug Content and pH

The content of FZ in the studied gel microemulsions ranged between 97.55 ± 0.48% and 102.05 ± 0.67% (Table 3), the values being within the limits indicated by official compendia, including our national pharmacopoeia, which tolerates a ±3% deviation from the declared value for products containing 0.5% and more than 0.5% of a drug [63]. These results highlight the uniform distribution of FZ in the experimental chitosan-gelled microemulsions.

All experimental gel microemulsions showed a weak acidity, indicated by slightly low pH values, from 4.50 ± 0.017 to 4.63 ± 0.025, with insignificant differences between values (Table 3). It is of note that, in comparison with the pH values of the fluconazole-loaded microemulsions (3.47 ± 0.06 to 3.72 ± 0.03) presented in our previous study [56], the pH of the corresponding gelled MEs increased by about one unit after the addition of CTS as gelling agent as a result of the protonation of the free amino groups at the C-2 position of the d-glucosamine units in its structure; measured pH values indicate that the protonation was total. Based on these results, it can be suggested that the pH of the studied ME gel formulations ranged within the limits recommended by official standards for semisolid dermal preparations [63] and in the normal range of the skin pH so they would not be a determinant factor of potential skin irritation after their application on the skin.

#### 3.1.2. Rheological Analysis of Chitosan-Gelled Microemulsions Based on Essential Oils and Sucrose Esters Containing 2% FZ

The results of the steady-state flow (viscosimetric) test are shown in Figure 1 in the form of rheograms and viscosity curves, as well as in Table 4 and Table 5 which summarize the values of viscosity, thixotropy and other rheological parameters obtained by fitting the viscosimetric test data with different rheological models.

The linear dependence of the shear stress (*τ*) on the deformation rate (*γ*) and the viscosity (*η*) decrease with the increase of the deformation rate indicate that all studied microemulsions are non-Newtonian pseudoplastic materials (Figure 1).

Also, Figure 1 and Table 4 show that all chitosan-gelled microemulsions presented a slight thixotropy, more pronounced for the MEGEL-FZ-D1616-ORG 6% formulation and for the formulations with CIN and 10% oil phase.

The pseudoplastic thixotropic flow of the experimental microemulsion gels has been mathematically described with great precision by both power law (R > 0.99) and the Casson model (R > 0.96) (Table 5). As a result, it can be suggested that the flow mechanism of the gelled microemulsions was similar, although there were differences among the formulations regarding the thixotropy degree.

The Herschel–Bulkley model accurately described the system’s viscosity variation according to the shear rate, which is shown by close to 1 values for the correlation coefficient (0.9644–0.9931) (Table 5). The viscosity of the experimental microemulsion gels ranged from 0.437 to 0.541 Pa·s (Table 4), falling within the specific range values for semisolid preparations. Small viscosity differences can be observed among the systems containing the same sucrose ester and the same concentration of the oily phase (i.e., 0.489 Pa·s for the MEGEL-FZ-D1616-CIN 6% formulation, 0.442 Pa·s for the MEGEL-FZ-D1616-ORG 6% and 0.477 Pa·s for the MEGEL-FZ-D1616-CLV 6% formulation). Additionally, there was a slight increase in viscosity with increasing the oil phase concentration from 6% to 10% for the formulations containing the same essential oil and the same surfactant. Similarly, small differences in viscosity values presented in the systems containing the oil phase at the same concentration (6% or 10%) but a different essential oil and surfactant.

The regression analysis of the viscosimetric steady-state flow test data allowed the calculation of two rheological parameters, namely the consistency index, *K*, and the flow index, *n*, which can be used to evaluate the influence of the formulation variables on the rheological properties of chitosan-gelled FZ microemulsions.

The consistency index (*K*), correlated with the gel viscosity at a shear rate of 1 s^−1^, is an indicator of its spreadability during its application on the skin. From Table 5 it can be observed that the highest *K* values (4.44 and 4.236) were calculated for two of the formulations containing CIN (MEGEL-FZ-D1616-CIN 10% and MEGEL-FZ- D1216-CIN 10%) and the lowest values (2.861 and 2.878) were obtained for two of the D1616-based microemulsions (MEGEL-FZ-D1616-ORG 10% and MEGEL-FZ-D1616-CLV 6%). The other formulations showed consistency values ranging from 3.336 to 4.086.

The values of the flow index *n*, less than 1 in all cases, confirmed the pseudoplasticity of the studied systems (Table 5). However, the pseudoplastic character was moderate (*n* > 0.5) with small differences between formulations, the *n* values varying in a narrow range (0.551–0.606). The highest values of the flow index (0.600–0.606), indicating the lowest pseudoplasticity, were calculated for microemulsion gel formulations containing 6% oily phase, except for MEGEL-FZ-D1616-CIN 6%, as well as for the MEGEL-FZ-D1616-ORG 10% formulation.

The data of the spreadability test were used to obtain the extensometric profiles of the 2% fluconazole-loaded microemulsions based on essential oils and sucrose esters and gelled with CTS (Figure 2).

For all studied formulations it can be observed that: (i) the spreadability increased progressively with the increase of the applied weight; (ii) all the obtained extensometric curves were similar; and (iii) they exhibited good spreadability, demonstrated by the obtained high extensometric values.

The highest spreadability values, which also generated superimposed extensometric profiles, were calculated for all applied weights in the case of the formulations containing ORG, except for the MEGEL-FZ-D1216-ORG 10% formulation, the spreading capacity of which was lower (being most likely correlated with higher viscosity). The formulations MEGEL-FZ-D1216-CIN 10% and MEGEL-FZ-D1616-CLV 10% had the lowest spreading capacity, which can be attributed to their increased viscosity.

### 3.2. In Vitro Drug Release and Skin Permeation Studies

#### 3.2.1. In Vitro Release of Fluconazole

Figure 3 shows the effect of the formulation variables (type and concentration of the essential oil, type of the sucrose ester) on the cumulative amounts of FZ released from the experimental MEGELs through the hydrophilic synthetic polysulfone membrane in the receptor medium (physiological saline solution with 60% ethanol) after 8 h of testing.

From Figure 3a, it can be observed that large amounts of FZ (89.88–99.37%) were released from all tested formulations. The highest cumulative drug percentages released after 5 or 6 h of testing (99.19% and 99.37%, respectively) were obtained for the formulations containing ORG and sucrose palmitate. These formulations were followed by the microemulsions based on CIN (MEGEL-FZ-D1616-CIN 10%, MEGEL-FZ-D1216-CIN 6%, MEGEL-FZ-D1616-CIN 6%, MEGEL-FZ -D1216-CIN 10%), from which the maximum cumulative FZ amount released after 8 h of testing was 98.74%, 98.65%, 97.47% and 97.73%. Additionally, the formulation MEGEL-FZ-D1216-ORG 10% released 97.56% FZ after 6 h of testing. A slightly lower cumulative FZ amount (94.70% and 91.27%) was released after 8 h from the microemulsion gels based on CLV and sucrose palmitate, while the MEGEL-FZ-D1216-ORG 6% formulation released the lowest cumulative percentage of FZ (89.88%) after 3 h of testing. The cumulative release kinetic profiles obtained for the studied formulations were similar and the plateau effect observed for some of the formulations (Figure 3a) corresponded to the depletion of FZ from the donor compartment, which led to a decrease of the concentration gradient along the membrane and to a reduced penetration rate.

The very high percentages of released FZ from the experimental formulations highlight the high capacity of the vehicle to release the drug, confirming the performance of the formulations.

Table 6 shows the values of the FZ permeation parameters (steady-state flux, *J_ss_*, and the permeability coefficient, *K_p_*) and the FZ release parameters (release rate, *k*, and diffusion coefficient, *D*) from the vehicle through the synthetic membrane, calculated based on the obtained experimental data.

It was found that the FZ transfer through the synthetic membrane was the most intense from the microemulsion gel based on ORG and sucrose laurate with 6% oily phase, which produced the highest value of the steady-state flux (975.89 ± 8.60 μg/cm^2^/h), although it did not release the largest cumulative amount of FZ (Figure 3b and Table 6). Additionally, large membrane flux values (ranging from 712.42 ± 9.94 μg/cm^2^/h to 808.51 ± 2.80 μg/cm^2^/h) were produced by three of the formulations containing sucrose palmitate, namely MEGEL-FZ-D1616-CIN 10%, MEGEL-FZ-D1616-ORG 10% and MEGEL-FZ-D1616-ORG 6% (Table 6). Of the other three microemulsion gels based on CIN (MEGEL-FZ-D1616-CIN 6%, MEGEL-FZ-D1216-CIN 6% and MEGEL-FZ-D1216-CIN 10%) and from the formulation MEGEL-FZ-D1616-ORG 10%, the FZ membrane transport was approximately 1.5 times lower compared to that from the MEGEL-FZ-D1616-ORG 6% formulation. The smallest flux values, indicating the lowest FZ transfer through the membrane, were produced by the systems based on CLV and D1616 (Figure 3b and Table 6).

The formulations based on D1616 and clove or cinnamon essential oil, as well as the MEGEL-FZ-D1216-CIN 10% formulation, showed a lag time (the *x*-axis intercept at the linear portion of the curves for permeated cumulative drug amount versus time under steady-state conditions), but having values of less than one hour is considered a normal feature of this type of curve.

To evaluate the in vitro drug release profiles of FZ from MEGELs, the experimental data were analyzed using four kinetic models and the fitting results are listed in Table 7.

As shown, the release profile of FZ from the MEGELs through the synthetic membrane could be best described by the Korsmeyer–Peppas model, which yielded higher determination coefficient values (R^2^ range 0.8956–0.9928). Furthermore, the values of diffusion exponent, *n*, varied in the range of 0.6768 to 0.9843, suggesting an anomalous non-Fickian transport (both diffusional and relaxational transport) in case of all MEGELs based on oregano and cinnamon essential oil, except the MEGEL-FZ-D1616-CIN 10% formulation, and also the MEGELs based on CLV, for which the *n* values were superior to 1, supporting a non-Fickian super case II drug transport mechanism.

#### 3.2.2. In Vitro Skin Permeation of Fluconazole

The specific permeation profiles of FZ from the experimental microemulsion gels through pig skin illustrated in Figure 4a show that significantly higher FZ cumulative percentages were released from most of the systems based on oregano and cinnamon essential oil (*p* < 0.05) compared to the formulations containing CLV: 91.10% (MEGEL-FZ-D1216-ORG 10%), 87.80% (MEGEL-FZ-D1616-ORG 10%), 87.06% (MEGEL-D1216-CIN 10%), 83.03% (MEGEL-FZ-D1216-ORG 6%), 77.61% (MEGEL-FZ-D1216-CIN 6%) and 72.27% (MEGEL-FZ-D1216-CIN 10%) versus 59.26% (MEGEL-FZ-D1616-CLV 10%) and 48.06% (MEGEL-FZ-D1616-CLV 6%). The maximum cumulative percentage of FZ released from the formulations containing sucrose palmitate with 6% oily phase, MEGEL-FZ-D1616-ORG 6% and MEGEL-FZ-D1616-CIN 6%, was close, respectively lower than that produced by the microemulsions containing CLV and the same surfactant (57.93% and 37.81%, respectively).

By comparing the dermato-pharmacokinetic profile of the FZ from microemulsion gels through porcine skin (Figure 4b) and the steady-state flux values calculated based on the in vitro data (Table 8), it can be seen that the highest amounts of FZ were transferred through the porcine skin from three formulations containing 10% oily phase, namely MEGEL-FZ-D1616-CIN 10%, MEGEL-FZ-D1216-SCR 10% and MEGEL-FZ-D1616-ORG 10%.

In contrast, all formulations based on D1616 and 6% of the oily phase (MEGEL-FZ-D1616-CLV 6%, MEGEL-FZ-D1616-ORG 6% and MEGEL-FZ-D1616-CIN 6%) produced the smallest values of the transcutaneous flux. The permeation rate of FZ from the microemulsion gels based on D1216 and ORG was slightly lower than that from the formulation with D1616, MEGEL-FZ-D1616-ORG 10%. Additionally, from Table 8 it can be seen that the FZ flux through the skin from the vehicle increased with the increase of the essential oil concentration (from 3% to 5%) for each of the five studied ternary systems.

In addition, the transcutaneous permeation profiles of FZ from the formulations based on D1616 and cinnamon or clove essential oil also showed a lag time with higher values for MEGEL-FZ-D1616-CIN 10%, MEGEL-FZ-D1616-CLV 6% and MEGEL-FZ-D1616-CLV 10% (0.93 ± 1.15 h, 1.56 ± 1.69 h and 1.33 ± 0.94 h, respectively) (Table 8). However, lag time values did not vary in all cases as would be expected, namely a longer lag time for a slow diffusion. For example, the MEGEL-FZ-D1616-CIN 10% formulation that led to the fastest FZ transfer through the skin showed a lag time of about 1 h, unlike the MEGEL-FZ-D1616-CIN 6% formulation from which FZ permeated at the lowest rate but after an insignificant lag time (0.19 ± 0.38 h).

### 3.3. Evaluation of In Vitro Antifungal Activity of the Experimental Gel Microemulsions

The results of the antifungal activity evaluation of the chitosan-gelled microemulsions based on volatile oils and sucrose esters containing 2% FZ are shown in Table 9 and illustrated in Figure 5 compared to the control formulations (drug-free). This was done in order to quantify/highlight the antifungal effect of FZ and also of the other gel microemulsion components (essential oil, sucrose ester and CTS).

As expected, all evaluated preparations, with or without FZ, inhibited fungal growth; the antimycotic activity increased with the increasing of the of essential oil concentration in the formulation for both control preparations and FZ-loaded formulations. Synergism between FZ and the essential oil regarding the antifungal effect is also highlighted, the diameter of the inhibition zone for the tested preparations being about 1.4–2.1 times greater than that of the control preparation.

The largest diameter of the inhibition zone, indicating the most pronounced antimycotic effect, was presented by the formulation MEGEL-FZ-D1216-ORG 10% (Table 9), closely followed by the formulations based on CIN and sucrose palmitate or laurate with 10% oily phase (5% essential oil), the two formulations based on CLV and sucrose palmitate and by the two formulations based on ORG and palmitate or sucrose laurate with 10% and 6% oily phase, respectively. In contrast, the other microemulsion gels containing 6% oily phase based on cinnamon or oregano essential oil inhibited fungal growth on a significantly lower area (*p* < 0.05), possessing lower antimycotic activity (Table 9 and Figure 5).

## 4. Discussion

The non-Newtonian pseudoplastic behavior is desirable in the case of topical pharmaceutical formulations to prevent the mobility of the dispersed phase under low shear stress because it is important that these systems flow freely when sheared, exhibiting low viscosity at high shear rates; however, these changes are reversible after a certain rest period, delaying the coalescence or cremation of the systems. The decrease in viscosity with the increase in shear rate is associated with the destruction of the entanglements of the polymer chains in the three-dimensional gel structure followed by the polymer chain alignment in the direction of the applied stress, which is responsible for the decrease in viscosity under these conditions.

The differences among the formulations regarding the viscosity, consistency and pseudoplasticity are most likely attributed to the structural changes which occur in the gel microemulsions due to the combined effects of formulation variables (the type and concentration of the essential oil, the nature of the sucrose ester). The polarity and different chemical composition of the essential oil (the main components being eugenol in CLV, eugenol and cinnamic aldehyde in CIN, carvacrol and thymol in ORG) and the length of the alkyl chain in the sucrose ester structure will change the surfactant’s solubility in the two microemulsion phases and its phase behavior. Consequently, both the arrangement of the surfactant molecules at the oil–water interface (implicitly the curvature and fluidity of the interfacial film) and the micellization are affected. Additionally, possible interactions between molecules of different microemulsion components (water, surfactant, cosurfactant and terpenes of essential oil) and the cationic hydrophilic polymer (CTS) may also be considered; these interactions may affect the gelling ability of CTS.

The results of the extensometric measurements were consistent with those of the steady-state flow test, indicating that the experimental formulations with lower consistency and viscosity exhibited a higher spreading capacity. Similarly, the differences between the studied formulations regarding their spreadability can be explained based on the same factors discussed for the viscosimetric assay.

The differences in the FZ transfer through the synthetic membrane from the studied gel microemulsions can be attributed to the combined effects of several factors, such as: (a) the drug solubility in the microemulsion components; (b) the FZ distribution among the three phases (oily internal phase, aqueous external phase and surfactant micelles); and (c) the proportion of the microemulsion components. Numerous previous studies have shown that the drug release from an L/H microemulsion is limited by the diffusion of the active compound from the oily and micellar phases in the external aqueous phase from where the drug molecules are available to be released. Moreover, the drug transfer from the micellar phase can cause a partial break of the micelle followed by the solubilization of the surfactant molecules in the aqueous phase of the microemulsion. In addition, due to the fact that the drug transfer from the micelles to the aqueous phase is much faster than that from the oil phase to the aqueous phase, decreasing the drug concentration in the aqueous external phase (attributed to the drug permeation through the membrane) determines, first, the drug transport from the micellar phase to the aqueous one and, consequently, the altering of the micelle structure. Therefore, the micelle rupture leads to increased drug solubility in the aqueous phase of the microemulsion, which represents the driving force for an increased release [64].

Analyzing the contribution of the above-mentioned factors to the FZ transfer from the studied chitosan-gelled microemulsions, the following considerations can be made. First, the FZ solubility in the essential oil makes an important contribution, as the highest flux values were obtained in the case of the formulations based on ORG (with the exception of MEGEL-FZ-D1216-ORG 10%), in which FZ presented the highest solubility. Second, the sucrose ester solubility/affinity in the oil phase and for the aqueous phase was also relevant, determining, on the one hand, the amount of FZ dissolved in the lipophilic phase and, on the other hand, the formation of surfactant micelles in the aqueous phase and the FZ diffusion in this phase. Since D1216 was found to be more water soluble and, therefore, less soluble in the oil phase than D1616 [56], it is distributed predominantly in the aqueous phase in the form of micelles, favoring the FZ diffusion from the oil phase by micellar solubilization, which increases the availability of the drug molecules to diffuse into the aqueous phase. However, sucrose palmitate, which is more soluble in the oily phase, is distributed in the aqueous phase as micelles to a lesser extent but increases the FZ solubility in the oily phase, which increases the concentration gradient of FZ between the two phases (oily and micellar), favoring the diffusion of the drug molecules in the aqueous phase. Third, the effect of the oily phase concentration in the formulation was evident especially in the case of the microemulsions containing D1616 (ester with higher solubility in oil). Thus, the increase of the oily phase proportion from 6% to 10% led to a decrease of the FZ flux from the systems containing oregano and clove essential oil (essential oils in which the FZ solubility was high), respectively, and to an increase of the FZ flux from the formulations with CIN in which the FZ solubility was low.

Other potential factors responsible for the differences between the studied formulations regarding the release of FZ through the synthetic membrane would be their viscosity/consistency. It is known that a lower viscosity of the topical formulation allows an easier drug diffusion through the vehicle than a higher viscosity due to the inverse relationship between the viscosity and the ability of the formulation to release the drug.

The two-step release mechanism based on polymer relaxation and diffusion phenomena, described by the Korsmeyer–Peppas model, was not observed for the studied MEGELs, the FZ profiles illustrated in Figure 3 indicating a controlled drug release throughout the experiment. This behavior was previously reported by other studies [65], the authors proposing a quasi-Fickian drug diffusion mechanism for such systems. In addition, it is considered that the drug release from microemulsion systems is a complex process also involving a drug transfer across liquid/liquid interfaces [66,67].

The results of the in vitro permeation test of FZ through the pig skin revealed the ability of the microemulsified systems based on essential oils and sucrose esters to increase the percutaneous penetration of FZ; on the other hand, they indicate the influence of several factors, mainly the physicochemical properties of the essential oils and sucrose esters on the FZ percutaneous penetration.

Although the *S_mix_* concentration was the same in all the studied formulations, the surfactant/cosurfactant ratio was different (1:1.5 for the systems *(CIN/IPM)-(D1616/IPA)*-water and *(ORG/IPM)-(D1616/IPA)*-water and 1:2 for the systems (*CIN/IPM*)-*(D1216/IPA)*-water, (*ORG/IPM*)-*(D1616/IPA)*-water and (*CLV/IPM*)-*(D1616/IPA)*-water); thus, the concentration of the two components in the microemulsions was different, namely: 18% sucrose ester and 27% isopropyl alcohol, and 15% sucrose ester and 30% isopropyl alcohol, respectively. Consequently, the percutaneous permeation of the drug was governed by both the concentration gradient and other factors, such as the composition and concentration of the essential oil, and the type and concentration of the surfactant.

It can be suggested that the composition and concentration of the essential oil were the most important formulation variables based on the following results: (i) except for the formulation MEGEL-FZ-D1616-ORG 6%, all the systems containing oregano and cinnamon essential oil presented a faster FZ transfer through porcine skin than those based on CLV; and (ii) the first three formulations in the hierarchy based on the flux values contained either cinnamon or oregano essential oil, and the oily phase concentration was 10% (corresponding to 5% essential oil). The last three formulations in the same hierarchy contained 6% oily phase (corresponding to 3% essential oil).

These results highlight the fact that oregano and cinnamon essential oils in concentration of 3% and especially 5% were more effective in promoting the FZ percutaneous penetration than CLV. The effect of the ORG can be attributed to its lower boiling point than that of CLV (239 °C versus 251 °C) and its higher dissolution ability of FZ. In the case of the CIN, its lower dissolution ability of FZ was most likely counterbalanced by its higher volatility due to its lower boiling point compared to that of the CLV (194–234 °C versus 251 °C). The suggested mechanisms by which the essential oils act as skin penetration enhancers are as follows: (a) interaction of the terpenes from their composition with the lipids of the corneous layer, altering its barrier property; and (b) increasing the coefficient partition corneous layer/vehicle of the drug, increasing its solubility in the lipids of the corneous layer. Additionally, the influence of the boiling points of the essential oils on their interaction with the corneous lipids was also discussed [68].

Furthermore, for the results interpretation, the synergistic effect of the sucrose ester with that of the essential oil on FZ percutaneous permeation should be also taken into account. Thus, it can be concluded that the type of surfactant and, to a lesser extent, its concentration in the formulation influenced the ability of the two sucrose esters to enhance drug skin penetration. Previous studies [43,69,70,71] have indicated that D1216 is a more effective penetration enhancer than D1616 for various drugs with moderate or low hydrophilicity, including FZ. However, this difference was not always found throughout our study, most likely due to the effect of the essential oil, which was predominant at the 10% oily phase concentration (corresponding to 5% essential oil). Thus, two of the microemulsions that produced the highest flux values, MEGEL-FZ-D1616-CIN 10% and MEGEL-FZ-D1616-ORG 10%, contained D1616 but also essential oil in higher concentration (5%); however, a high flux value was produced by MEGEL-FZ-D1616-CIN 10%, a formulation based on D1216 and CIN (5%). In contrast, the lower ability as penetration enhancer of D1616 compared to D1216 was found in the formulations with 6% oil phase (3% essential oil); MEGEL-FZ-D1616-CIN 6% and MEGEL-FZ-D1616-ORG 6% formulations produced FZ flux values about two times lower than the same microemulsion gels with D1216 (MEGEL-FZ-D1216-CIN 6% and MEGEL-FZ-D1216-ORG 6%).

The variation of the sucrose ester concentration in the experimental microemulsion gel composition, determined by the different surfactant/cosurfactant ratio in *S_mix_*, may be another factor responsible for the unexpected results of our study. For example, the higher flux value produced by the MEGEL-FZ-D1616-CIN 10% formulation compared to that of the MEGEL-FZ-D1216-CIN 10% formulation can be attributed also to higher D1616 concentration than that of D1216 (18% versus 15%), although the latter is more potent as penetration enhancer.

The increased antimycotic effect of the plain microemulsion gels, mainly due to the essential oil present in the composition, is worth noting. The antifungal properties of the three essential oils used in the study (essential oils of cinnamon, oregano, clove) have already been reported in the literature by many authors [72,73,74].

Because essential oils contain various volatile compounds, including terpenoids, esters, aldehydes, ketones, acids and alcohols, it is difficult to attribute the antifungal activity to a single compound or class of compounds. In fact, the antifungal effects of the essential oils are due to several synergistically acting compounds; but, in practice, these effects are attributed to the majority component/components of the volatile oil. Thus, the major components responsible for the antifungal properties of the essential oils used in this study are: cinnamaldehyde for CIN; carvacrol and thymol for ORG; and eugenol and cinnamaldehyde for CLV [72,73,74].

The results of the antifungal activity of control formulations based on essential oils and sucrose esters are encouraging as they suggest their potential use as an effective alternative in clinical management of cutaneous/mucosal candidiasis resistant to currently used antifungal drugs.

## 5. Conclusions

Novel, biocompatible, chitosan-gelled microemulsions based on essential oils (cinnamon, oregano or clove essential oil) and sucrose ester (sucrose palmitate or sucrose laurate) for effective topical delivery of antifungal drug fluconazole were successfully developed. These formulations were characterized in terms of drug content, pH, rheological properties (viscosity, rheological behavior and spreadability), in vitro drug release and skin permeation and antifungal activity.

The obtained results indicated that the composition and concentration of the essential oil, as well as the type and concentration of the sucrose ester in the fluconazole microemulsion gel formulations, significantly affected the evaluated properties. The in vitro FZ permeation through porcine skin revealed, by the obtained dermatological-pharmacokinetic parameters, the ability of the experimental microemulsion gels to increase drug percutaneous penetration.

According to the obtained results, the best formulations for topical FZ release were considered to be the following microemulsion gels: MEGEL-FZ-D1616-CIN 10%, MEGEL-FZ-D1216-CIN 10% and MEGEL-FZ-D1616-ORG 10%, containing a 10% mixture of essential oil and isopropyl myristate (in a 1:1 ratio) as the oily phase, 45% *S_mix_* (sucrose palmitate-isopropyl alcohol 1:1.5, sucrose laurate-isopropyl alcohol 1:2 and sucrose palmitate-isopropyl alcohol 1:2) as surfactant–cosurfactant, 1.9% CTS as gelling agent and 2% acetic acid as the aqueous phase. These formulations produced the highest flux values, as well as large diameters of the fungal growth inhibition zone (*Candida albicans* spp.). The formulations MEGEL-FZ-D1216-ORG 10%, MEGEL-FZ-D1216-ORG 6% and MEGEL-FZ-D1616-CLV 10% presented a more pronounced or similar antifungal effect.

It can be concluded that these biocompatible gel microemulsions based on essential oils and natural sucrose ester surfactants and gelled with CTS could be used as vehicles for simultaneous topical administration of both groups of biologically active compounds (essential oil and FZ). However, further advanced preclinical and clinical tests on the microstructure of the emulsified systems, gel structure, components’ solubility in the liquid phases of the system, textural profile, stability, innocuity, skin permeation and in vitro and in vivo therapeutic efficacy of active components are required. If the studies are successful, the proposed optimized pharmaceutical formulations can be implemented in therapy to offer the benefits of simultaneous therapeutic effects of the active ingredients (natural and synthetic) associated with the favorable properties of CTS and sucrose esters at dermal application.

## Figures and Tables

**Figure 1 pharmaceutics-14-00075-f001:**
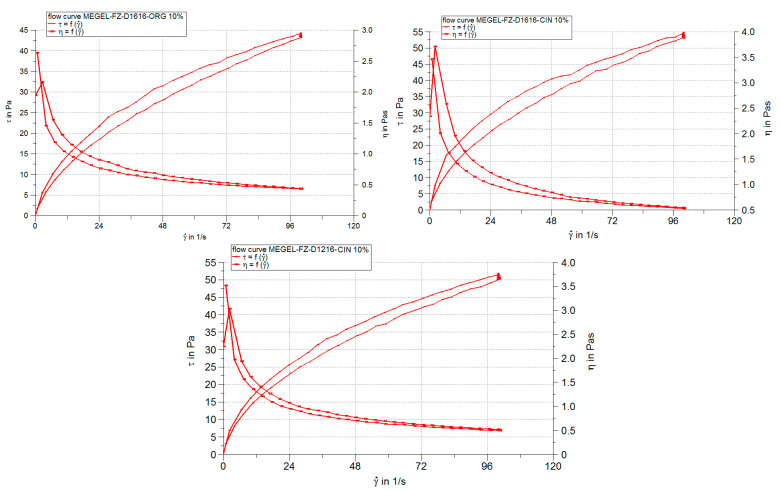
Rheograms and viscosity curves of some chitosan-gelled microemulsions containing 2% FZ and based on different essential oils and sucrose esters.

**Figure 2 pharmaceutics-14-00075-f002:**
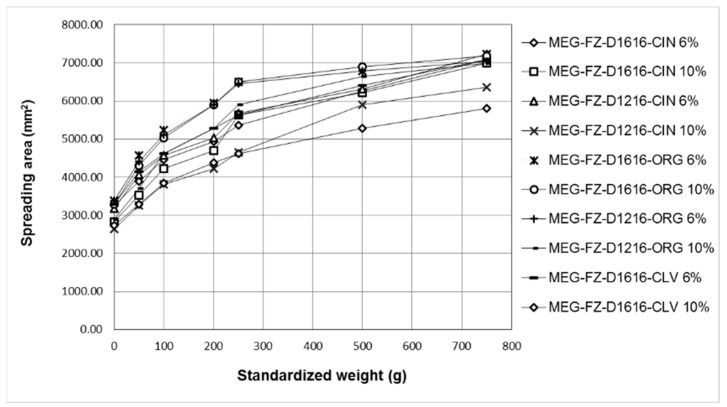
The spreadability profiles of the gel microemulsions based on sucrose esters and essential oils containing 2% fluconazole.

**Figure 3 pharmaceutics-14-00075-f003:**
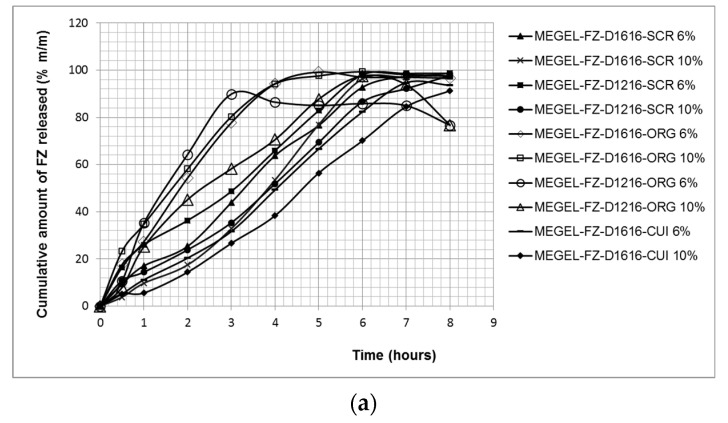
The cumulative kinetic profiles of FZ release from investigated gel MEs through synthetic membrane (mean ± SD, *n* = 5): cumulative amount of fluconazole released (in % (**a**), and in μg/cm^2^ (**b**)) versus time.

**Figure 4 pharmaceutics-14-00075-f004:**
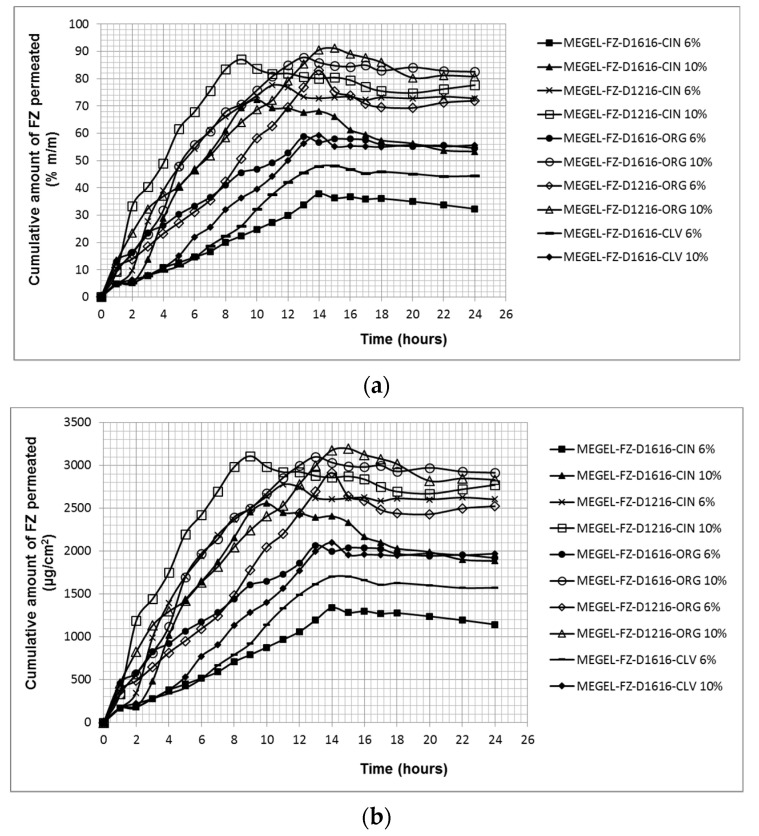
The permeation profiles of FZ through porcine skin from microemulsions based on essential oils and sucrose esters gelled with chitosan: cumulative amount of fluconazole permeated (in % (**a**), and in μg/cm^2^ (**b**)) versus time.

**Figure 5 pharmaceutics-14-00075-f005:**
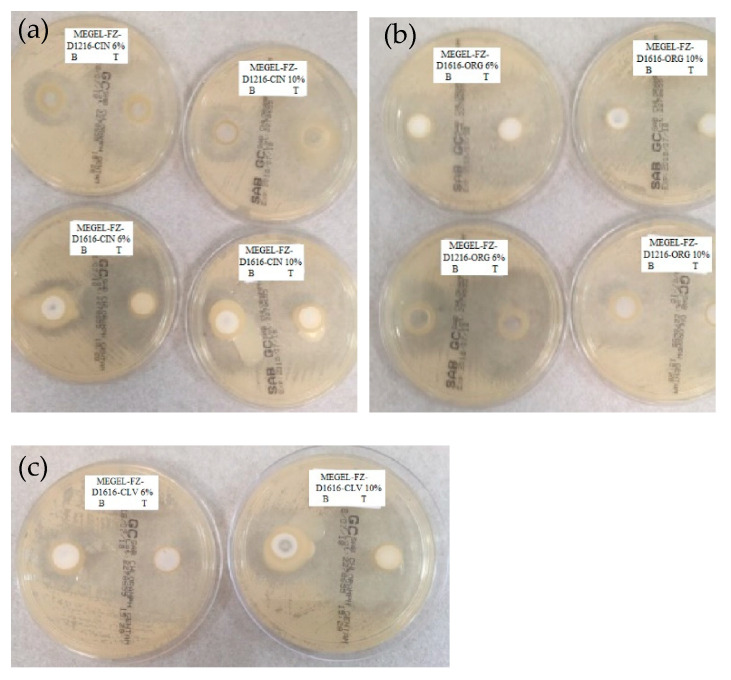
Fungal growth inhibition of *Candida albicans* in the presence of experimental gel microemulsions containing 2% fluconazole and cinnamon essential oil (**a**), oregano essential oil (**b**) or clove essential oil (**c**).

**Table 1 pharmaceutics-14-00075-t001:** Composition (%, *w/w*) of fluconazole-loaded gel microemulsions based on sucrose palmitate, essential oil and chitosan.

Gel Microemulsion Components	ME-FZ-D1616-CIN 6%	ME-FZ-D1616-CIN 10%	ME-FZ-D1616-ORG 6%	ME-FZ-D1616-ORG 10%	ME-FZ-D1616-CLV 6%	ME-FZ-D1616-CLV10%
Fluconazole	2.0	2.0	2.0	2.0	2.0	2.0
Cinnamon essential oil	3.0	5.0	-	-	-	-
Oregano essential oil	-	-	3.0	5.0	-	-
Clove essential oil	-	-	-	-	3.0	5.0
Isopropyl myristate	3.0	5.0	3.0	5.0	3.0	5.0
Sucrose palmitate– isopropanol (1:1.5)	45.0	45.0	-	-	-	-
Sucrose palmitate– isopropanol (1:2)	-	-	45.0	45.0	45.0	45.0
Chitosan	1.9	1.9	1.9	1.9	1.9	1.9
2% acetic acid solution	47.0	43.0	47.0	43.0	47.0	43.0

**Table 2 pharmaceutics-14-00075-t002:** Composition (%, *w/w*) of fluconazole-loaded gel microemulsions based on sucrose laurate, essential oil and chitosan.

Gel Microemulsion Components	ME-FZ-D1216-CIN 6%	ME-FZ-D1216-CIN 10%	ME-FZ-D1216-ORG 6%	ME-FZ-D1216-ORG 10%
Fluconazole	2.0	2.0	2.0	2.0
Cinnamon essential oil	3.0	5.0	-	-
Oregano essential oil	-	-	3.0	5.0
Isopropyl myristate	3.0	5.0	3.0	5.0
Sucrose laurate– isopropanol (1:2)	45.0	45.0	-	-
Sucrose laurate– isopropanol (1:1.5)	-	-	45.0	45.0
Chitosan	1.9	1.9	1.9	1.9
2% acetic acid solution	47.0	43.0	47.0	43.0

**Table 3 pharmaceutics-14-00075-t003:** Fluconazole content and pH of the chitosan-gelled microemulsions based on essential oils and sucrose esters.

Formulation Code	Drug Content (%)	pH
MEGEL-FZ-D1616-CIN 6%	100.62 ± 0.53	4.50 ± 0.017
MEGEL-FZ-D1616-CIN 10%	99.15 ± 0.27	4.50 ± 0.011
MEGEL-FZ-D1216-CIN 6%	97.55 ± 0.48	4.52 ± 0.005
MEGEL-FZ-D1216-CIN 10%	98.43 ± 0.72	4.50 ± 0.004
MEGEL-FZ-D1616-ORG 6%	101.34 ± 0.96	4.51 ± 0.010
MEGEL-FZ-D1616-ORG 10%	99.75 ± 0.44	4.52 ± 0.016
MEGEL-FZ-D1216-ORG 6%	100.62 ± 0.81	4.50 ± 0.019
MEGEL-FZ-D1216-ORG 10%	102.05 ± 0.67	4.58 ± 0.047
MEGEL-FZ-D1616-CLV 6%	98.85 ± 0.29	4.63 ± 0.025
MEGEL-FZ-D1616-CLV 10%	101.87 ± 0.65	4.52 ± 0.013

**Table 4 pharmaceutics-14-00075-t004:** Rheological properties of the experimental microemulsion gels containing 2% FZ at 23 °C.

Formulation Code	Apparent Viscosity * (Pa∙s)	Thixotropy (Pa/s)
MEGEL-FZ-D1616-CIN 6%	0.489	623.1
MEGEL-FZ-D1616-CIN 10%	0.539	741.7
MEGEL-FZ-D1216-CIN 6%	0.463	383.3
MEGEL-FZ-D1216-CIN 10%	0.508	778.3
MEGEL-FZ-D1616-ORG 6%	0.477	804.5
MEGEL-FZ-D1616-ORG 10%	0.437	398.4
MEGEL-FZ-D1216-ORG 6%	0.461	447.5
MEGEL-FZ-D1216-ORG 10%	0.541	487.5
MEGEL-FZ-D1616-CLV 6%	0.442	425.4
MEGEL-FZ-D1616-CLV 10%	0.528	577.9

**η*–apparent viscosity (Pa·s) calculated at a shear rate of 0.8 1/s.

**Table 5 pharmaceutics-14-00075-t005:** Parameters of rheological models and determination coefficient values specific to the respective models obtained in regression analysis of viscosimetric test results.

Formulation Code	Parameters of Rheological Model		Determination Coefficient Specific to Rheological Model		
	*K*	*n*	R_Power law_	R_Casson_	R_Herschel–Bulkley_
MEGEL-FZ-D1616-CIN 6%	3.359	0.594	0.9962	0.9774	0.982
MEGEL-FZ-D1616-CIN 10%	4.44	0.561	0.9929	0.9662	0.9644
MEGEL-FZ-D1216-CIN 6%	3.036	0.604	0.998	0.9801	0.9931
MEGEL-FZ-D1216-CIN 10%	4.236	0.551	0.9964	0.9748	0.9913
MEGEL-FZ-D1616-ORG 6%	3.179	0.601	0.9969	0.978	0.9831
MEGEL-FZ-D1616-ORG 10%	2.861	0.606	0.9954	0.9769	0.9841
MEGEL-FZ-D1216-ORG 6%	3.047	0.600	0.9981	0.9867	0.9877
MEGEL-FZ-D1216-ORG 10%	3.962	0.580	0.9974	0.9771	0.9905
MEGEL-FZ-D1616-CLV 6%	2.878	0.604	0.997	0.9798	0.9924
MEGEL-FZ-D1616-CLV 10%	4.086	0.568	0.9968	0.9751	0.9912

*K–*consistency index (Pas); *n–*flow index.

**Table 6 pharmaceutics-14-00075-t006:** Specific release parameters of FZ from the microemulsion gels through synthetic membrane.

Formulation Code	*J_ss_* (μg/cm^2^/h)	*K_P_* × 10^−6^ (cm/h)	*t_L_* (h)
MEGEL-FZ-D1616-CIN 6%	560.56 ± 1.79	280.28 ± 0.93	0.12 ± 0.85
MEGEL-FZ-D1616-CIN 10%	712.42 ± 9.94	356.21 ± 6.45	0.75 ± 1.13
MEGEL-FZ-D1216-CIN 6%	521.46 ± 1.21	260.73 ± 0.66	-
MEGEL-FZ-D1216-CIN 10%	548.38 ± 3.48	274.19 ± 2.52	0.60 ± 1.04
MEGEL-FZ-D1616-ORG 6%	808.51 ± 2.80	404.26 ± 1.73	-
MEGEL-FZ-D1616-ORG10%	741.87 ± 1.24	370.94 ± 0.68	-
MEGEL-FZ-D1216-ORG 6%	975.89 ± 8.60	487.95 ± 7.34	-
MEGEL-FZ-D1216-ORG 10%	512.71 ± 2.03	256.36 ± 1.89	-
MEGEL-FZ-D1616-CLV 6%	504.36 ± 2.51	252.18 ± 3.08	0.38 ± 0.62
MEGEL-FZ-D1616-CLV 10%	464.08 ± 5.15	232.04 ± 3.46	0.79 ± 0.86

**Table 7 pharmaceutics-14-00075-t007:** Results of kinetic analysis of FZ in vitro permeability data from the gel microemulsions through synthetic membrane.

Formulation Code	Zero Order		First Order		Higuchi		Korsmeyer–Peppas		
	*K*_0_ (μg/h)	R^2^	*K*_1_ (h^−1^)	R^2^	*K_H_* (h^−0.5^)	R^2^	*K_P_* (h^−*n*^)	*n*	R^2^
MEGEL-FZ-D1616-CIN 6%	13.353	0.9713	0.4915	0.9164	32.738	0.8967	1.2203	0.9107	0.9819
MEGEL-FZ-D1616-CIN 10%	14.298	0.9642	0.5614	0.8834	31.374	0.8143	0.949	1.2653	0.9885
MEGEL-FZ-D1216-CIN 6%	12.851	0.9569	0.5981	0.8903	34.814	0.9436	1.402	0.6768	0.9827
MEGEL-FZ-D1216-CIN 10%	12.875	0.9862	0.4234	0.8797	30.364	0.8712	1.207	0.8062	0.9607
MEGEL-FZ-D1616-ORG 6%	12.268	0.8088	0.7620	0.8724	39.154	0.926	1.4765	0.8	0.9897
MEGEL-FZ-D1616-ORG 10%	11.798	0.8032	0.6461	0.9029	39.945	0.9391	1.5607	0.6608	0.9919
MEGEL-FZ-D1216-ORG 6%	9.4754	0.6128	0.4041	0.8331	35.729	0.7958	1.4356	0.8954	0.8956
MEGEL-FZ-D1216-ORG 10%	11.492	0.8287	0.4373	0.8495	33.892	0.9042	1.2938	0.9843	0.9616
MEGEL-FZ-D1616-CLV 6%	13.023	0.9834	0.3202	0.7972	29.131	0.8379	1.0285	1.0796	0.9928
MEGEL-FZ-D1616-CLV 10%	12.139	0.9864	0.2133	0.8400	25.540	0.7916	0.9058	1.096	0.939

*K*_0_: Zero-order release constant; *K*_1_: First-order release constant; *K_H_*: Higuchi release constant; *K_P_*: Korsmeyer–Peppas release constant; *n*: diffusion coefficient in the Korsmeyer–Peppas model; R^2^: determination coefficient.

**Table 8 pharmaceutics-14-00075-t008:** Specific permeation parameters of FZ from the chitosan-gelled microemulsions through porcine skin.

Formulation Code	*J_ss_* (μg/cm^2^/h)	*K_P_* × 10^−6^ (cm/h)	*t_L_* (h)
MEGEL-FZ-D1616-CIN 6%	91.96 ± 9.88	45.98 ± 5.31	0.19 ± 0.38
MEGEL-FZ-D1616-CIN 10%	302.4 ± 3.55	151.2 ± 0.87	0.93 ± 1.15
MEGEL-FZ-D1216-CIN 6%	214.6 ± 4.91	107.3 ± 2.56	-
MEGEL-FZ-D1216-CIN 10%	288.4 ± 9.73	144.2 ± 5.82	-
MEGEL-FZ-D1616-ORG 6%	127.5 ± 6.55	63.75 ± 2.68	-
MEGEL-FZ-D1616-ORG10%	239.6 ± 8.35	119.8 ± 5.44	-
MEGEL-FZ-D1216-ORG 6%	190.6 ± 4.25	95.3 ± 2.14	-
MEGEL-FZ-D1216-ORG 10%	197.0 ± 7.22	98.5 ± 3.68	-
MEGEL-FZ-D1616-CLV 6%	135.8 ± 1.45	67.9 ± 0.55	1.56 ± 1.69
MEGEL-FZ-D1616-CLV 10%	165.4 ± 0.75	82.7 ± 0.49	1.33 ± 0.94

**Table 9 pharmaceutics-14-00075-t009:** Antifungal activity of the chitosan-gelled microemulsions.

Formulation Code	Diameter of the Inhibition Zone (mm)	
	Control	Sample
MEGEL-FZ-D1616-CIN 6%	24 ± 0.51	48 ± 0.39
MEGEL-FZ-D1616-CIN 10%	42 ± 0.37	59 ± 0.27
MEGEL-FZ-D1216-CIN 6%	26 ± 0.19	48 ± 0.51
MEGEL-FZ-D1216-CIN 10%	28 ± 0.13	58 ± 0.43
MEGEL-FZ-D1616-ORG 6%	30 ± 0.24	48 ± 0.33
MEGEL-FZ-D1616-ORG 10%	38 ± 0.17	56 ± 0.52
MEGEL-FZ-D1216-ORG 6%	34 ± 0.42	58 ± 0.28
MEGEL-FZ-D1216-ORG 10%	41 ± 0.38	62 ± 0.61
MEGEL-FZ-D1616-CLV 6%	26 ± 0.11	54 ± 0.17
MEGEL-FZ-D1616-CLV 10%	28 ± 0.45	58 ± 0.36

## Data Availability

All data available are reported in the article.

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
