# Peer review of "New, Biocompatible, Chitosan-Gelled Microemulsions Based on Essential Oils and Sucrose Esters as Nanocarriers for Topical Delivery of Fluconazole"

_pharmaceutics, 2021, doi:10.3390/pharmaceutics14010075_

Round 1

Reviewer 1 Report

My report:

Manuscript ID: pharmaceutics-1481335

Title: New biocompatible chitosan-gelled microemulsions based on essential oils and sucrose esters as nanocarriers for topical delivery of fluconazole

Authors: Lavinia Vlaia, Ioana Olariu, Ana Maria MuÅ£, Georgeta Coneac, VicenÅ£iu Vlaia, Dan Florin Anghel, Monica Elisabeta Maxim, Gabriela Stângă, Amadeus Dobrescu, Maria Suciu, Zoltan Szabadai, Dumitru Lupuleasa

Overview and general recommendation:

The study aimed to improve the adhesiveness of respective fluconazole microemulsions using chitosan as gellator, and to evaluate the formulation variables effect (composition and concentration of essential oil, sucrose ester structure) on the gel microemulsions (MEGELs) properties. The routine work for evaluation of all MEGELs regarding drug content, pH, rheological behavior, viscosity, spreadability, in vitro drug release and skin permeation, and antifungal activity was carried out.

Overall, I found the manuscript is well designed and written. The methods are clearly described, and the results are supported by enough references and statistical analysis. However, I ask that the authors specifically address the following minor recommendations before publication.

Comments to the authors:

  1. Line 199, 252, 672, correct the typo mistake in “physicochemicaly”, “con-plate geometry”, “occure”, …...
  2. The authors should revise the abbreviations used such as Fluconazole and other polymers and then use it.
  3. Visualization of most figures is necessary.

Author Response

Comment 1:

Line 199, 252, 672, correct the typo mistake in “physicochemicaly”, “con-plate geometry”, “occure”, …...

 Answer:

Please note that we have corrected the typo mistakes.

Comment 2:

The authors should revise the abbreviations used such as Fluconazole and other polymers and then use it.

Answer:

Please note that we have revised the abbreviations used for Fluconazole, chitosan, cinnamon essential oil, oregano essential oil, clove essential oil, sucrose laurate and sucrose palmitate, and used them through the manuscript as such (FZ, CTS, CIN, ORG, CLV, D1216 and D1616 respectively).

Comment 3:

Visualization of most figures is necessary.

 Answer:

Please note that we have tried to improve the quality of the figures presented in the manuscript.

Reviewer 2 Report

The authors present data on a new biocompatible chitosan-gelled micro-emulsion based on sucrose esters and essential oils as a topical drug delivery system for fluconazole.

Although the idea is appealing, there are items that has to be addressed before publication.

  • The authors should more clearly state what kind of effect they want to achieve, local or systemic, e.g. the role of Stratum corneum accumulation of the drug. Furthermore, the interpretation of skin permeation is quite different in both cases. If a systemic effect is intended, what rational is behind the testing of the antifungal activity of the micro-emulsions. On the other hand, if a local effect is primarily wanted, why should skin permeation be high.
  • One rationale for using chitosan was its mucoadhesivity. However, when the carriers are applied to the skin there is no mucus layer to interact with.
  • Release studies as well as permeation studies are lacking an appropriate control, e.g. hydro-alcoholic solution of the API.
  • The authors use physiological saline solution containing 60 % ethanol as the receptor fluid. Are they sure that there is no precipitation when this receptor fluid comes into contact with the micro-emulsions in the pores of the separating membrane. This is an essential issue when discussing release profiles.
  • What are the values for LOD and LOQ of the used fluconazole assay? Is it sufficiently sensitive for the permeation studies?
  • Release profiles have been analyzed for their kinetic using different mathematical approaches. However, it could have been clearly expected that the power law model (Korsmeyer-Peppas) would give the best fit because it comprises all other models which only differ in the value of n. Thus a fit to the power law would have been sufficient.
  • The evaluation of the permeation profiles using the same approach must be partly erroneous because permeation always is accompanied by a certain lag time which is not reflected by either of the mathematical models.
  • The authors calculate from the release and permeation profiles so called permeation as well as release parameters. The only difference seems to be that the evaluation was done either on a released/permeated amount vs. time or released/permeated amount vs. square root of time. But this does not refer to different processes but to different kinetics.
  • The discussion is two lengthy and should be shortened addressing only relevant items.

Typos:

Line 123: it must read 10-200 nm instead of 10-200 µm

Reviewer 3 Report

The article entitled “New biocompatible chitosan-gelled microemulsions based on essential oils and sucrose esters as nanocarriers for topical delivery of fluconazole” is an interesting research on the possible use of hydrogel system for fluconazole delivery.

Some remarks on drug release experiments:

The presence of ethanol in the dissolution test is not allowed because the drug release curves deviate from the true value (see article “Ethanol effects on drug release from Verapamil Meltrex®, an innovative melt extruded formulation” International Journal of Pharmaceutics 368 (2009) 72–75).

Why was ethanol put in the dissolving medium?

Releases should be done without ethanol.

In figure 3 it is not clear the percentage of release (in ordinate permeate ug/cm2?).

The application of the equations is therefore incorrect and false of the drug release trend.

Author Response

Comment 1:

The presence of ethanol in the dissolution test is not allowed because the drug release curves deviate from the true value (see article “Ethanol effects on drug release from Verapamil Meltrex®, an innovative melt extruded formulation” International Journal of Pharmaceutics 368 (2009) 72–75).

Why was ethanol put in the dissolving medium?

Releases should be done without ethanol.

 Answer:

In vitro drug release test of topical dermatological drug products is a valuable and important performance test for which the European and USA regulatory authorities published several documents describing the test methodology (FDA Guidance for Industry SUPAC-SS. Nonsterile Semisolid Dosage Forms, Scale-Up and Postapproval Changes: Chemistry, Manufacturing, and Controls. In Vitro Release Testing and In Vivo Bioequivalence Documentation. U.S. Department of Health and Human Services, Food and Drug Administration, Center for Drug Evaluation and Research. June 1997; United States Pharmacopeial Convention, USP/NF 36/31, General chapter <1724>; Ueda CT, Shah VP, Derdzinski K, Ewing G, Flynn G, Maibach H, Marques M, Rytting H, Shaw S, Thakker K, Yacobi A. Topical and Transdermal Drug Products: Stimuli to the Revision Process. Pharmacopeial Forum. 2009; 35(3): 750-764; EMA/ CHMP/QWP/708282/2018. Draft Guideline on quality and equivalence of topical products). All these documents indicate that over the entire course of the experiment drug release must take place into a diffusional sink. To achieve sink conditions, the receptor medium must have a high capacity to dissolve or carry away the drug, and the receptor media should not exceed 10% of Cs (drug solubility in the releasing matrix) at the end of the test. For sparingly water soluble drugs (including fluconazole), the use of a hydroalcoholic medium as a receptor phase is essential to increase drug solubility and to maintain sink conditions.   

Comment 2: In figure 3 it is not clear the percentage of release (in ordinate permeate ug/cm2?). The application of the equations is therefore incorrect and false of the drug release trend.

Answer:

Please note that we have supplemented figures 3 and 4 with graphs illustrating the percentage of cumulative amount of fluconazole released over time.

Round 2

Reviewer 2 Report

The authors refined the text to make more clear that the intention of their study was to provide an antimycotic preparation that acts locally within the skin.

However, there are still some issues in this context not completely resolved.

In order to prove that the API reaches the site of action the determination of penetration profiles would have been the adequate test procedure.

The permeation still points to the amount of drug which is systemically available and not intended in the context of the present study.

Moreover, the authors claim an additional antimycotic effect through the addition of the used excipients.

If the site of action lies within the skin, e.g. living epidermis, it has to be proven that these compounds co-penetrate with the API fluconazole. Otherwise the in vitro test for antifungal activity does not help to support the hypothesis of improved effect through the excipients. Especially in case of the polymer chitosan it seems to be quite unlikely that this substance penetrated crosses the epidermal barrier.

Answer to Comment 3:

Sound scientific studies require reasonable control experiments. 

The authors claim that they present an innovative formulation concept. If so, results from other published formulations concepts (even though also named microemulsion) cannot serve as a 

The authors claim that they present an innovative formulation concept. If so, results from other published formulations concepts (even though also named microemulsion) cannot serve as proof of concept and  hypothetical control.